# A Comprehensive Review of Fragile X Syndrome and Fragile X Premutation Associated Conditions in Africa

**DOI:** 10.3390/genes15060683

**Published:** 2024-05-25

**Authors:** Chioma N. P. Mbachu, Ikechukwu Innocent Mbachu, Randi Hagerman

**Affiliations:** 1Department of Paediatrics, Faculty of Medicine, College of Health Sciences, Nnamdi Azikiwe University, Nnewi Campus, Nnewi 435101, Nigeria; 2MIND Institute, University of California Davis, Sacramento, CA 95817, USA; 3Department of Obstetrics and Gynaecology, Faculty of Medicine, College of Health Sciences, Nnamdi Azikiwe University, Nnewi Campus, Nnewi 435101, Nigeria; ii.mbachu@unizik.edu.ng; 4Department of Pediatrics, University of California Davis Health, Sacramento, CA 95817, USA

**Keywords:** fragile X syndrome, fragile X premutation-associated conditions, treatment, Africa

## Abstract

Fragile X syndrome (FXS) is a genetic disorder caused by a mutation in the fragile X messenger ribonucleoprotein 1 (*FMR1*) gene and known to be a leading cause of inherited intellectual disability globally. It results in a range of intellectual, developmental, and behavioral problems. Fragile X premutation-associated conditions (FXPAC), caused by a smaller CGG expansion (55 to 200 CGG repeats) in the *FMR1* gene, are linked to other conditions that increase morbidity and mortality for affected persons. Limited research has been conducted on the burden, characteristics, diagnosis, and management of these conditions in Africa. This comprehensive review provides an overview of the current literature on FXS and FXPAC in Africa. The issues addressed include epidemiology, clinical features, discrimination against affected persons, limited awareness and research, and poor access to resources, including genetic services and treatment programs. This paper provides an in-depth analysis of the existing worldwide data for the diagnosis and treatment of fragile X disorders. This review will improve the understanding of FXS and FXPAC in Africa by incorporating existing knowledge, identifying research gaps, and potential topics for future research to enhance the well-being of individuals and families affected by FXS and FXPAC.

## 1. Introduction

Fragile X syndrome (FXS), first reported by Martin and Bell [1] in 1943, is an X-linked dominant genetic disorder known to be the most common cause of hereditary intellectual and developmental disability globally [2]. It is caused by a mutation that involves an abnormal expansion of a CGG repeat in the 5′ region of the fragile X messenger ribonucleoprotein 1 (*FMR1*) gene, leading to the silencing of the gene and a deficiency in the production of the *FMR1* protein (FMRP), which regulates numerous processes in the human body [2,3,4,5,6]. FXS is caused by the full mutation (>200 CGG repeats), which silences the gene through methylation so there is little or no FMRP produced. FXS is typically characterized by cognitive deficits, behavioral and emotional problems including autism, and physical features which vary among different patients but may include a long face, large ears, large testicles after puberty (macroorchidism), and hyperextensible finger joints [7,8,9]. Normal alleles have less than 45 CGG repeats; the intermediate zone, also known as the ‘gray zone’, is between 45 and 54 CGG repeats and may expand in future generations. Premutation disorders have 55 to 200 CGG repeats and the gene is not silenced [7,10]. Usually, individuals with the premutation have normal intellectual abilities, although shyness, social anxiety and sometimes depression are common. A carrier male with the premutation will pass on the premutation to all his daughters who receive his only X chromosome, but all his sons will not have the mutation because they receive his Y chromosome. A carrier female with the premutation, however, will pass on the gene mutation to 50% of her children and expansion to a full mutation occurs with increasing frequency depending on the size of the premutation. Females with over 90 to 100 CGG repeats will expand to a full mutation every time that mutated X is passed on [7].

The molecular basis of FXS is the expansion of the CGG repeat sequence at the 5’ untranslated region (UTR) of a highly conserved *FMR1* gene, with 17 exons and a length of approximately 38 kb, at Xq27.3 [7,11]. This expansion leads to hypermethylation of repeat sequences and adjacent promoter regions, which decreases gene transcription so little, or no mRNA is produced, and this is called a full mutation (FM)-causing FXS [12]. FXS females demonstrate widely variable symptoms and are not as severely affected as FXS males because they have two X chromosomes, and the normal one is producing FMRP. There is an inverse relationship between disease severity in FXS females and the activation ratio (AR; meaning the percentage of cells that have the normal X as the active X). The higher the AR, the more FMRP is produced and the higher the IQ [7]. Some individuals with an FM may have some cells unmethylated or some cells with the premutation and this is termed mosaicism, either from size or methylation mosaicism. If there is a lack of complete methylation or several cells with the premutation, then more FMRP is produced, and cognitive abilities are higher, especially in the boys with mosaicism [13].

Fragile X premutation-associated conditions (FXPAC), also known as premutation disorders, include the Fragile X-associated Tremor/Ataxia syndrome (FXTAS), which is a neurodegenerative disorder seen in older carriers; Fragile X-associated Primary Ovarian Insufficiency (FXPOI) meaning menopause before age 40; and Fragile X-associated neuropsychiatric disorders (FXAND), which includes anxiety, depression, insomnia, and other neuropsychiatric disorders [14].

Each disorder in the premutation range is distinct and has its own associated clinical characteristics; they may occur over the entire life span of the affected individual.

For example, in infancy, problems with vision and motor functions are encountered; in childhood, it may be anxiety, seizure disorders, neurobehavioral problems, learning issues, social deficits, autism spectrum disorders and ADHD. Young adults can also have neurobehavioral problems and, in mid-adulthood, sometimes endocrine or immune problems may develop, especially in females, including FXPOI, hypothyroidism, insomnia, chronic fatigue and fibromyalgia. Also in adulthood, psychiatric and neurological problems may be overt, and there can be significant features of FXAND such as anxiety, depression, psychosis and sleep disorders. The elderly may exhibit neurodegenerative symptoms associated with FXTAS—tremors, ataxia, cognitive decline, parkinsonism, autonomic dysfunction and memory problems [14].

The global prevalence of FXS is estimated at 1 in 5000 males and 1 in 4000–8000 females [15]. A large population screening for FXS among neonates in the United States reported a prevalence rate of 1:5161 among male neonates [16]. In Québec, Canada, a screening of over 24,000 neonates found a prevalence rate of 1 in 6209 among male participants, while FXS was not found among the female neonates screened [17]. In a systematic review and meta-analysis by Hunter et al. [18], the premutation is estimated to occur at a rate of 1 in 250–800 males and 1 in 110–270 females, similar to reports from a book by Tassone and Hall as editors [19]. Most of these findings are from America and Europe.

FXS and FXPAC have not been adequately studied in Africa, and there are sparse data on the burden, diagnosis, and management of these conditions in the continent.

In this comprehensive review, we aim to explore the prevalence of FXS and FXPAC in Africa, highlighting the challenges and gaps in research, as well as potential strategies for improving diagnosis and management of these conditions in the region.

## 2. Epidemiology of FXS in Africa

The burden of FXS and FXPAC in Africa has received relatively little attention in the scientific literature. Research on FXS in Africa is sparse due to limited awareness, diagnostic services, and personnel, with most studies focusing on other continents such as North America and Europe [20,21,22]. However, there are some data that suggest that the prevalence of FXS in Africa may be under-reported due to a lack of awareness and diagnostic services. In addition, the genetic diversity of African populations may affect the prevalence and presentation of FXS in this region [23,24].

It is estimated that FXS affects a similar proportion of the population in Africa as in other parts of the world, with a prevalence of approximately 1 in 5000 males and 1 in 4000–8000 females [15]. A large, 20-year retrospective population review conducted by Essop et al. [20] among over 2000 individuals with intellectual disability in urban South Africa found a 5.2% prevalence rate of FXS among African participants. Another population study on intellectually disabled institutionalized adolescent and adult males in rural South Africa reported a slightly higher prevalence rate of 6.1% for FXS among participants studied [25]. Although both studies were conducted in the same country, the populations studied were different. The former was conducted on those with intellectual disability in the general population, while the latter was among institutionalized males. This could have accounted for the mild disparity in rates.

In Egypt, North Africa, Meguid et al. [26] found a rate of 6.4% (16/250) with FXS among intellectually disabled in-school male children and adolescents. This was almost like a study by Goldman and Jenkins [25] among intellectually disabled institutionalized males in South Africa.

In contrast, Kamga et al. [27] in Yaounde, Cameroon, Central Africa, after cascade testing, reported a 4 in 19 (21%) prevalence rate among boys with intellectual disability in a large extended family. This high prevalence rate could be explained by the founder effect (participants tested were descendants of the founder).

Latunji [28] in Ibadan, southwest Nigeria, reported a 6.83% (4/69) prevalence rate of FXS among intellectually disabled children and adults who were enrolled in special schools or living in homes. This rate was like the findings by Meguid et al. [26] in Egypt despite the variations in the population studied, sample size and diagnostic method. Latunji’s [28] study used both cytogenetic and molecular methods in the diagnosis, whereas Meguid et al. [26] used a molecular method (PCR). However, these studies are limited in scope and may not accurately reflect the true prevalence of FXS in Africa. In contrast to studies from Africa, a systematic review and meta-analysis by Hunter et al. [18], among mostly European and Americans, found that 2.4% (178/7475) of the population with intellectual disability had FXS. This rate is lower than all reports from the African subregion, suggesting that FXS may be more prevalent in the region.

### 2.1. FXPAC

FXPAC includes a group of disorders caused by the premutation, which leads to a higher level of mRNA than normal. The elevated mRNA levels in premutation carriers causes RNA toxicity, leading to clinical involvement, so this is very different from those with an FM and FXS [3,7]. These disorders are in the premutation range, implying that affected persons have unmethylated, CGG repeats in the range of 55 to 200 and the level of FMRP is usually in the normal range [14,15,29]. They were previously thought to be carriers who were largely unaffected, but further research over the years has revealed that, actually, premutation carriers may have a plethora of symptoms and signs that largely cause morbidity and mortality if left unattended [14]. The premutation (PM) carriers with greater than 120 CGG repeats have a mild reduction in FMRP levels, which can lead to mildly prominent ears and hyperextensibility of finger joints similar to those with FXS [7,14,30]. The pathogenesis of PM disorders in male and female carriers can be explained by markedly elevated *FMR1* mRNA, directly related to the length of the repeats, leading to activation of cellular stress pathways, sequestration of proteins important for neuronal function, mitochondrial dysfunction and other RNA-mediated toxicity [14,31,32,33].

Alleles in the ‘gray zone’, also known as intermediate alleles, with 45 to 54 CGG repeats may occasionally expand to the premutation range or even the full mutation after the second generation [34].

### 2.2. Studies on Fragile X Premutation Associated Conditions in Africa

While there is limited research on FXS in Africa, even less is known about FXPAC, such as FXTAS, FXPOI and the most recently named FXAND. These conditions do not cause intellectual disability but can still have a significant impact on affected individuals.

In a large study of unrelated participants in Johannesburg, South Africa, the prevalence of the premutation was noted to be 0.54% (12/2239) among participants [20]. This prevalence rate is slightly higher than the rate of about 0.4% documented in the systematic review by Hunter et al. [18], which did not include African studies and other global estimates of PM. Hunter et al. found 0.1–0.3% in males and 0.3–0.4% in females in the general population [35,36]. Another study among 148 unrelated, institutionalized South African males with intellectual disability found the prevalence of premutation to be zero (0) among participants [25]. This is not an unexpected finding considering the fact that participants in this study all had intellectual disability, which is common in the FM but not in those with the PM [3,37,38,39,40]. A study in Egypt reported a prevalence of FXPOI of 40% in female carriers, which is also higher than global estimates [22]. Although there are limited data on the prevalence of these conditions in Africa, these studies suggest that FXPAC may indeed be more common in certain African populations than previously thought, highlighting the need for further research.

### 2.3. FXPOI

Premature ovarian insufficiency (POI) is a diverse condition typified by a premature decline in the function of the ovaries before 40 years old [41]. It affects approximately 3.5% of women globally [42]. It is also known as hypergonadotropic hypogonadism and characteristic features include low estrogen levels, elevated gonadotropins (FSH > 30 U/I), and either primary or secondary amenorrhea of 4 to 6 months [43].

POI remains idiopathic as the cause is still unknown; however, some factors have been implicated as possible mechanisms including genetic, metabolic, environmental toxins, autoimmune disorders, pelvic surgery, cancer treatment with chemotherapy and/or radiotherapy [41]. Genetics has been implicated in playing a major role in the pathogenesis of POI [44,45]. POI has been associated mainly with the X chromosome and autosomal mutations. Turner syndrome (45, X), with a complete loss of one X chromosome, caused by haploinsufficiency of inactivated X-linked genes, is known to be a predominant genetic cause of syndromic POI [46,47]. A notable report in the history of FXPAC is the report of premature menopause (FXPOI) by conference attendees who were women in their early thirties and mothers of boys with FXS. This event happened in 1986 before the discovery of the *FMR1* gene [14]. Now, the *FMR1* premutation is recognized to be the most common genetic etiologic factor for POI [48], with FXPOI occurring in about 1 in every 5 women with the premutation [7,14,29,49]. The prevalence of the PM in women is estimated at 1 in 150, with over 1 million women in the United States known to have the premutation allele [50]. These data on the prevalence of FXPOI remain to be seen, particularly in settings that have diagnostic challenges [51]. Reports have shown that women who knew their premutation status and also had symptoms of POI experienced delays in receiving a diagnosis of FXPOI. However, ‘diagnostic delays’ were more pronounced in women who were unaware of their premutation status [52]. These women usually present with either primary or secondary infertility, hot flashes, osteoporosis, and other associated features of low estrogen levels [7,49].

Another study demonstrated increased prevalence of premature menopause among premutation carrier women who had either an average or normal IQ, which is expected in those with the PM [53]. Women who carry the premutation alleles have increased risk of developing FXPOI [7,14,49,54]. There is an increased risk for women with these highly unstable, usually unmethylated premutation alleles to give birth to offspring in the FM if they conceive. There is a bell-shaped relationship between FXPOI in PM carriers and the CGG repeat number, with the greatest incidence of FXPOI found among those with CGG repeats between 85 and 100 [48,55,56]. Premutation carriers occur in 7 out of every 100 of sporadic cases of POI, while the prevalence increases with a familial history and about 21 in 100 women have the premutation with familial POI [41]. In contrast, a lower prevalence rate of approximately 1% has been reported for POI in the general population [56,57]. However, there is still a slim chance of fertility in women with FXPOI; thus, the name was changed from premature ovarian failure (POF) to POI a few years ago [46].

Both premutation carriers and healthcare personnel have demonstrated a great need to identify risk and predictive markers for FXPOI, particularly because of the increased prevalence of women who are premutation carriers from several screening studies [58,59,60,61,62,63,64,65].

There is limited research on the prevalence of FXPOI in Africa. Essop and Krause [20] reported an estimated 33% prevalence of FXPOI (1 in 3 females tested who had a PM) in POI referrals in Johannesburg, South Africa, in their 20-year review. This is significantly higher than the rates reported by previous authors, suggesting that FXPOI and other FXPAC may indeed be more prevalent than presumed [14].

The treatment of FXPOI includes hormonal replacement and psychotherapy if needed and this is discussed in detail elsewhere [66].

### 2.4. FXTAS

FXTAS, as its name implies, is a neurodegenerative syndrome associated with the premutation and characterized by tremor and ataxia; specific clinical and radiological criteria make one qualify for a diagnosis of this disorder [14,67,68].

FXTAS, like other conditions in the FXPAC spectrum, is caused by trinucleotide expansion of 55 to 200 CGG repeats in the *FMR1* gene [69]. The pathogenesis revolves around markedly elevated *FMR1* mRNA, cellular stress, increase in reactive oxygen species (ROI), mitochondrial dysfunction and further RNA toxicity, leading to CGG-binding protein sequestration with the excess mRNA and repeat-associated non-AUG-initiated (RAN) translation, leading to production of a toxic protein, FMRpolyG [14,32,33].

The characteristic pathological findings consist of intranuclear inclusions that are positive for ubiquitin. These inclusions are seen in neurons, astrocytes, and Purkinje cells in the brain, spinal cord, and autonomic ganglia throughout the body [70,71,72].

This neurodegenerative disorder usually starts in older carriers above 50 years, and it is more common and more severe in males than females [73,74]. Women are less susceptible to FXTAS since about half of their neurons and glial cells produce the normal X chromosome instead of the X chromosome with the premutation. This results in reduced toxicity [75].

FXTAS is so diverse that it may present with a variety of features beyond the tremor and ataxia including parkinsonism, executive function deficits, memory loss, cognitive deficits with eventual dementia, brain atrophy, white matter disease characterized by hyperintensities on MRI involving the middle cerebellar peduncles (MCP sign), the splenium and periventricular areas. In addition, motor and autonomic dysfunction, neuropsychiatric symptoms like anxiety, irritability, apathy and depression and peripheral neuropathy are common in those with FXTAS [69,76,77,78,79,80,81,82,83,84,85].

The conclusive diagnostic criteria for this medical condition require the presence of one major clinical symptom (either gait ataxia or intention tremor) and one major radiological finding (white matter lesions in the MCP on MRI) in an affected individual [67,68,69]. Other secondary indicators such as neuropathy, executive function impairments, parkinsonism, overall brain atrophy, and damage to the splenium of the corpus callosum can be utilized to establish a Probable or Possible diagnosis of FXTAS when combined with an intention tremor or ataxia [67,69].

The prevalence of FXTAS is estimated at 40–75% in males compared to 13–16% in females [7,68].

This prevalence increases with increasing age and is said to affect almost 75% of male carriers by 80 years of age [68].

Some medications can improve tremors such as primidone or a beta (β) blocker, but ataxia is more difficult to treat. Psychiatric problems such as depression, anxiety or irritability can be treated with a selective serotonin reuptake inhibitor (SSRI). There is some evidence that memantine can improve memory or focus [67], but studies have not been carried out regarding treatment of dementia. Newer medications that can improve mitochondrial function, such as oral allopregnanolone or Anavex 2-73, will hopefully lead to greater improvement in this disorder [14].

There is limited research on FXTAS in Africa. In fact, from our literature search, no study was found on FXTAS, thus highlighting the need for awareness, testing, resources, and research in this area since FXTAS mimics various other neurological disorders.

### 2.5. FXAND

A more recent concept introduced an additional categorization called FXAND to specifically address the higher occurrence of mental health issues including anxiety, depression, ADHD, insomnia, chronic pain and chronic fatigue [75].

The conditions under the umbrella of FXAND can begin in childhood, including anxiety presenting as severe shyness and ADHD in both males and females with the premutation. Often anxiety can worsen in adulthood and can be accompanied by insomnia and depression. Chronic fatigue, chronic pain from migraines or muscle aches, or fibromyalgia are common. Autoimmune problems such as autoimmune thyroiditis can occur, but more severe problems such as lupus or multiple sclerosis have also been reported in 1 to 3% of carriers of the premutation [65].

Treatment of FXAND includes SSRIs as noted above, particularly for anxiety, depression, or obsessive-compulsive thinking. Avoidance of CNS toxins including alcohol, pesticides, illegal drugs, smoking, and excessive opioids can help alleviate symptoms and slow progression to FXTAS. In addition, exercise can be very helpful for anxiety and depression and chronic pain or fatigue. Therapy is always indicated for psychiatric problems and antioxidants may also help to relieve oxidative stress [86]. From our literature search, no specific study addressed FXAND in Africa. This review may serve to create awareness of the disorder in the African region.

### 2.6. Clinical and Behavioral Phenotypes of FXS

The cognitive abilities in those with FXS depend largely on FMRP levels. The level of FMRP is related to the presence of mosaicism in males and to the AR in females [87]. Once the CGG repeat is above 273 in the FM, whether methylated or unmethylated, there is no FMRP produced so it makes no difference if the CGG repeat number is 400 or 1000 [88].

Mildly low FMRP levels are associated with less severe symptoms, normal or borderline intelligence and less severe learning disabilities as in many females with the FM. In contrast, very low or absent FMRP levels lead to moderate to severe cognitive impairment and intellectual disability [11].

Production of FMRP also determines physical characteristics in affected individuals [11]. Characteristic facial features are seen in 8 out of 10 post pubertal men and these include a long face, prominent ears, and large testicles (macroorchidism) [87].

However, in young children, approximately 30 to 40% may not demonstrate typical facial features. A prominent forehead and hyperextensible finger joints are seen in most children. Enlarged testicles or macroorchidism do not occur until puberty. In 1991, Hagerman et al. [37] developed a checklist to aid clinicians to easily identify patients with FXS. This checklist contained 13 items (intellectual disability, hyperactivity, short attention span, tactile defensiveness, hand flapping, hand biting, poor eye contact, perseverative speech, hyperextensible metacarpophalangeal joints, large or prominent ears, large testicles, simian crease or Sydney line, and family history of intellectual disability). Five or more clinical criteria were associated with a much higher risk for FXS (Table 1).

A few years after Hagerman’s checklist was developed [37], Maes et al. [40] developed a 28-item checklist comprising 7 physical and 21 behavioral characteristics for the identification of patients in need of further referral for gene testing. Physical features listed include narrow and elongated face, high forehead, prominent lower jaw, large protruding ears, macroorchidism, hyperextensible finger joints and other hyperextensible joints. Behavioral features mentioned in the checklist are hyperactivity, sensory oversensitivity, impulsivity, being chaotic, shyness, gaze avoidance, being too helpful, approach-avoidance conflict, fearfulness, gaiety/cheerfulness, hypersensitivity for changes, hand-biting, stereotypic hand movement, flapping hands and arms, turning away the face, tactile defensiveness, rapid speed of language, being talkative, perseveration, echolalia, and imitation of language. It is pertinent to note that this checklist, while containing a lot of items, may be repetitive, and vary across cultural lines and co-morbidities like autism spectrum disorders.

A 15-item checklist adapted and modified from Hagerman’s [37] original 13-item checklist was developed by Indian researchers more than a decade later, to reflect the cultural variations in dysmorphology in FXS [39]. Items included physical (large ears, elongated face, broad forehead, large testicles, hyperextensible joints and simian crease) and neurological characteristics (intellectual disability, hyperactivity, gaze avoidance, positive family history of intellectual disability, hand biting, hand flapping, short attention span, tactile defensiveness and perseverative speech).

A meta-analysis by Lubala et al. [89] developed a concise seven-item universal clinical checklist for FXS (soft and velvety skin with redundancy of skin on the dorsum of the hand, flat feet, large and prominent ears, plantar crease, large testes in post pubertal boys, family history of intellectual disability and autistic-like behavior), based on findings from Europe and Asia because of insufficient data from Africa (Table 2). This lack of data from Africa prompted the recent development of another checklist by Lubala et al. [38] to reflect their findings in African subjects from three families in Congo, Central Africa (Table 2). This is likely the first attempt to phenotypically differentiate FXS in Africa from other cultures, bearing in mind cultural variations in clinical and behavioral phenotypes. Items mentioned in the physical features include elongated face, soft and velvety skin, and plantar crease, which were seen in all cases; large testicles, hyperextensible joints, simian crease and prominent ears, which were were seen in over two-thirds of cases; and about 14 cases had microcephaly, which is an unusual finding in FXS. The behavioral characteristics include severe intellectual disability, hyperactivity, short attention span and autistic-like features which were seen in all cases. This new checklist may be utilized albeit cautiously in resource-limited settings as a diagnostic aid for FXS and calls for further studies on clinical and behavioral phenotypes of FXS in Africa, with a population of multi-ethnocultural regions.

### 2.7. Challenges in Diagnosing FXS and FXPAC in Africa

One of the major challenges in diagnosing FXS and FXPAC in Africa is the lack of awareness among healthcare professionals and the general population about the condition. This has also been reported in other developing nations [59]. Many individuals with FXS may go undiagnosed or misdiagnosed due to lack of clinical expertise, leading to delays in receiving appropriate care and support.

Another challenge is the limited access to genetic testing, resources, inadequate healthcare infrastructure and specialized healthcare services in many African countries [20,21,22,90].

Genetic testing for FXS and associated disorders now employs more advanced molecular technologies like Southern blot and newer PCR-based methods compared to older methods of testing [91]. These are often not widely available, and healthcare facilities may not have the necessary expertise to diagnose and manage the condition effectively [20,21]. Many countries are advocating for newborn screening for FXS and there have been significant strides made in this regard [10,65,92]. Attempts have been made by a few researchers for the possibility of using clinical and behavioral phenotypes as a means of diagnosis in resource-poor settings with conflicting findings [38,90]. Lubala et al. [38] in Congo found clinical and behavioral phenotyping with checklists reported by Hagerman et al. [37], Maes et al. [40] and Guruju et al. [39] to be helpful. In contrast, Lumaka et al. [90] in Congo, Central Africa, noted that none of the 105 individuals assessed with checklists reported by previous authors had FXS after molecular testing, further highlighting the need for gene testing for suspected cases. The paucity of genetic services in Africa may result in a lack of accurate data on the prevalence of FXS in Africa and hinders efforts to improve diagnosis and treatment of the condition.

An additional challenge is stigma and discrimination against individuals and families with FXS or other intellectual disabilities [93]. This stigma from cultural beliefs may hinder early identification and testing because families and or communities may conceal affected individuals to prevent alienation by unaffected community members. Kamga et al. [94] in their qualitative study in Cameroon reported the experience of a mother who had two children with FXS. She was a descendant of the founder of a powerful royal family and despite this, they clearly experienced stigma, isolation, and name-calling. The disorder was said to have been a curse on the founder of the family which then transmitted to his descendants. Cultural beliefs about intellectual disabilities contribute to under-diagnosis and under-treatment of these conditions [93].

### 2.8. Genetic Testing for FXS and FXPAC in Africa

Genetic testing for FXS and FXPAC is essential for accurate diagnosis, genetic counseling, and management of these conditions. However, access to genetic testing in Africa is limited, with most centers located in urban regions [20], and there is a lack of awareness among some healthcare providers about the importance of testing for FXS and its associated conditions. Efforts to improve access to genetic testing and training for healthcare providers are needed to ensure that individuals with FXS and FXPAC receive the appropriate care and support.

### 2.9. Clinical Management of FXS and FXPAC in Africa

The clinical management of FXS and FXPAC should start from a detailed history taking, which should consider all important aspects of the history, including the developmental, nutritional, social history, drug history, allergy history and family history. After this, a thorough physical examination should be conducted on the individual. In the case of FXS, look for phenotypical features; in FXPAC, check for tell-tale signs relevant to the condition and other co-morbidities. For example, FXTAS patients may have ataxia and tremor, which can be detected by a neurological examination. The next step will be to undergo genetic testing if the patient has not had testing, and if it is available. Subsequently, proper treatment should be carried out.

The treatment of FXS/FXPAC can be pharmacological and non-pharmacological.

From the literature search, none of the African studies discussed the treatment or management of FXS/FXPAC in Africa [20,22,26,28,38,89,90]. Here, we discuss evidence-based treatments and supportive care for these conditions.

## 3. Clinical Management of FXS

Individuals with FXS typically enjoy a normal state of health, apart from frequently occurring ear infections in the first few years of life, seizures in 12–15% of cases, and connective tissue issues, such as joint dislocations, hernias and collapsable Eustachian tubes predisposing to otitis media resulting from the absence or insufficiency of FMRP [15].

Recurrent ear infections are prevalent in early childhood among individuals with FXS, affecting more than 50% of males with FXS [95]. This issue is likely attributed to a combination of factors including the shape of the face, collapsible Eustachian tubes, enlarged tonsils or adenoids, which may block the Eustachian tube causing recurrent infections. The treatment of recurrent otitis media infections in childhood includes the use of antibiotics for the acute infection in addition to insertion of pressure equalizing (PE) tubes. Language is boosted and significantly improves after the insertion of PE tubes [7].

### 3.1. Seizures

The prevalence of seizures in FXS is higher compared to the general population [95,96]. A wide variation exists in the results of prevalence from initial studies on FXS patients in genetics, epilepsy and neurology clinics compared with the community. The prevalence rate found in the genetics clinics was between 9% and 27%, epilepsy and neurology clinics reported 14% to 44%, while the community was 12% to 18%. This could have occurred because of a selection bias in the population studied [96]. FXS males are at a higher risk for seizures than females, with seizures occurring in 12% of males versus 3% in females [95,96]. However, the information at hand indicates the actual population prevalence of seizure incidence at 10% to 15% [7].

Many types of seizures do occur in FXS, although partial seizures are presumed to be the most frequently occurring type. However, generalized seizures are quite common; others are simple, complex partial and status seizures. An electroencephalogram (EEG) should be carried out in patients with possible seizures to help differentiate seizure type, e.g., benign Rolandic seizures vs. others.

Risk of autism spectrum disorder (ASD) increases in people with FXS and seizures. Similarly, the chances of seizure occurrence were higher in those who had FXS and co-occurring ASD [96]. Monotherapy with non-sedating and non-behaviorally aggravating antiseizure medications are preferred and achieve seizure control, with most patients being seizure-free. Medications are usually discontinued after 2 years without any seizure episode [7].

Preferred medications with minimal side effects are Keppra (levetiracetam) and oxcarbazepine (Trileptal), and Lamictal (lamotrigine) or Depakote (valproic acid) for refractory seizures [7].

#### Pharmacotherapy

There has been great advancement in the treatment of FXS. FXS currently has no cure and, as such, treatment aims to manage symptoms. Stimulants work well for ADHD symptoms after 5 years of age and guanfacine helps with hyperactivity in children younger than 5 years [97]. Anxiety is pervasive in both males and females with FXS and an SSRI such as sertraline can be helpful. A controlled trial of low-dose sertraline (2.5 to 5 mg per day) was carried out in children with FXS ages 2 to 6 years old and sertraline was superior to placebo in many developmental subtests on the Mullen Scales for Early Learning [98]. If aggression is problematic, then low doses of an atypical antipsychotic such as aripiprazole or risperidone have been helpful in children with FXS [97]. Currently, there are targeted treatments for FXS. These include metformin, which lowers the mTOR pathway that is elevated in FXS. Open-label studies of metformin have shown improvement in IQ in 2 adults with FXS treated with metformin for a year [99]. In an open-label study of young children with FXS ages 2 to 7 years old, metformin improved developmental testing on the Mullen Scales compared to previously published cognitive testing in children with FXS [100].

A controlled trial of metformin is currently underway in children and adults ages 6 to 40 years old and results will be available in the summer of 2024.

Cannabidiol (CBD) is a GABA agonist that also impacts the serotonin system, and it is anti-inflammatory. In a controlled trial in children ages 3 to 18 with FXS, CBD was efficacious in the primary outcome measure of social avoidance from the Aberrant Behavior Checklist, but only in those who were >90% methylated [101]. CBD was manufactured so there was no tetrahydrocannabinol (THC) and it was applied to the skin as a gel twice a day. It was very well tolerated, and caregiver reports demonstrated an improvement in aggression, anxiety and tantrums so that the families were able to take their children out in public, for instance, to restaurants without an outburst, so this improved their quality of life [101]. However, the FDA was concerned that those with less than 90% methylation or those who were mosaic did not demonstrate significant efficacy, so a second phase 3 trial is now being carried out at multiple international sites.

Perhaps the most exciting, targeted treatment for FXS is the Tetra drug called BPN14770 (Zitolamast), which is a phosphodiesterase 4D (PDE4D) inhibitor that inhibits the breakdown of cyclic adenosine monophosphate (cAMP), which is too low in FXS. So, this PDE4D allows cAMP to rise to normal levels which enhances neuronal connectivity. The results of a controlled trial in adults with FXS demonstrated efficacy by improving subtests on the National Institute of Health (NIH) toolbox including oral reading, picture vocabulary and overall Crystalized Intelligence Composite score [102]. This is an exciting result seeing cognition improve in adults, so currently both adolescent and adult studies are being carried out at multiple centers in a phase 3 trial.

### 3.2. Non-Pharmacological Therapy

Behavioral therapy is beneficial in patients with FXS in addition to speech and language therapy, occupational therapy and physical therapy in childhood, and this is usually received in a special education program through their school.

Additionally, there is a need for research on the efficacy of various treatment options for FXS in African populations, as well as the development of culturally appropriate interventions for affected individuals and their families. Guidance for behavioral interventions can be found in the literature and also on the National Fragile X Foundation website at www.fragileX.org, (accessed on 1 April 2024)

### 3.3. Conclusions

The burden of FXS and FXPAC in Africa, though important, is still under-studied and warrants further research and attention. Challenges in diagnosing and managing FXS in Africa include limited awareness, research, access to genetic testing and specialized healthcare services, and discrimination against affected persons.

These challenges need to be addressed to improve diagnosis, treatment, and support for individuals and families with FXS and FXPAC in the African continent.

### 3.4. Future Line of Study

Future research on FXS/FXPAC in Africa should focus on conducting large-scale epidemiological studies, developing culturally appropriate and acceptable screening tools, improving access to genetic testing and diagnostic services, raising awareness and education about the disorders even among health workers, and addressing stigma and discrimination against individuals with intellectual disabilities. Furthermore, there is a need for more research on genetic variability across different ethnicities and subregions in Africa and how this influences the burden and clinical picture of FXS/FXPAC. This will contribute to a deeper understanding of these disorders in the African region.

## Figures and Tables

**Table 1 genes-15-00683-t001:** Hagerman’s Fragile X checklist.

	Not Present (Score 0)	Borderline or Present in the Past (Score 1)	Definitely Present (Score 2)
**Intellectual disability**			
**Hyperactivity**			
**Short attention span**			
**Tactilely defensive**			
**Hand flapping**			
**Hand biting**			
**Poor eye contact**			
**Perseverative speech**			
**Hyperextensible metacarpophalangeal joints**			
**Large or prominent ears**			
**Large testicles**			
**Simian crease or Sydney line**			
**Family history of intellectual disabilities**			
**Total score**			

Adapted from Hagerman et al. [37].

**Table 2 genes-15-00683-t002:** Lubala et al. [38,89] clinical checklists for FXS.

Old Checklist	New Checklist	Not Present (Score 0)	Present (Score 1)
Soft and velvety skin with redundancy of skin on the dorsum of hand	Soft and velvety skin		
Flat feet	Elongated face *		
Large and prominent ears	Prominent ears		
Plantar crease	Plantar crease		
Large testes in post pubertal boys	Large testicles		
Family history of intellectual disabilities	Simian crease *		
Autistic-like behavior	Hyperextensible joints *		
	Microcephaly *		
	Severe intellectual disabilities		
	Hyperactivity *		
	Short attention span *		
	Autistic-like features		
Total score			

* = additions to the old checklist.

## Data Availability

No new data were created or analyzed in this study. Data sharing is not applicable to this article.

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
