# Peer review of "A Comprehensive Review of Fragile X Syndrome and Fragile X Premutation Associated Conditions in Africa"

_genes, 2024, doi:10.3390/genes15060683_

Round 1

Reviewer 1 Report

Comments and Suggestions for Authors

The manuscript is focusing on a very important topic with a significant impact on the research on genetic conditions particularly rare disorders. Lack of information on diverse populations has been a major concern of clinicians and the research community. So, I was very excited about reviewing this manuscript. 

Authors performed a comprehensive review and I appreciate the information related to the multiple studies conducted in Africa. However, the manuscript would have been more impactful if it was more focused on those specific studies. Similar studies in African Population related to some of the co-occurring conditions could have been mentioned in the manuscript. See below some additional comments: 

·       Fig 1 is the exact figure as Tassone et al. 2023. Even though the proper citation is included, I recommend removing the figure. Authors can include some text elaborating on the impact of the FXPAC across the lifespan. 

·       In section 2, epidemiology of FXS in Africa, authors mentioned the prevalence estimate of FXS in the general population and reported on studies in Africa that are mostly based on individuals with intellectual disability. 

It will be helpful to add studies conducted in other countries where the rate is also estimated based on individuals with intellectual disability. It will provide a better picture of similarities/differences in FXS rate between Africa and other parts of the world. 

·       The text in section 2.1., epidemiology of FXPAC in Africa, is not relevant to the title of the section. Similar pattern throughout the manuscript. It includes general information about fragile X syndrome or premutation but limited information specific to African populations or studies conducted in Africa (e.g., the section 3, lines 380-453, does not include information related to African population). 

The structure of the manuscript needs revision. In the introduction section include a brief literature review and then in the main text discuss information related to the African population. 

For example, in the section on "Clinical and Behavioral Phenotypes of FXS", authors can focus more on checklist by Lubala et al. Provide addition information on what is different in the new checklist, what is the implication and how it can be adopted in development of other checklists/ guidelines related to the African population. 

Comments on the Quality of English Language

The manuscript is well written. Here are a few minor suggestions:

 In many cases the sentences are too long or lack clarity. Please revise throughout the manuscript (e.g., Page 1: Lines 37-45, Page 2: Lines 59-61)

Some of the acronyms are defined multiple times. Please revise throughout the manuscript. 

Author Response

Response to Reviewer 1 Comments

2. Point-by-point response to Comments and Suggestions for Authors

Comment 1: The manuscript is focusing on a very important topic with a significant impact on the research on genetic conditions particularly rare disorders. Lack of information on diverse populations has been a major concern of clinicians and the research community. So, I was very excited about reviewing this manuscript. 

Authors performed a comprehensive review and I appreciate the information related to the multiple studies conducted in Africa. However, the manuscript would have been more impactful if it was more focused on those specific studies. Similar studies in African Population related to some of the co-occurring conditions could have been mentioned in the manuscript. See below some additional comments: 

Response 1:  Thank you for pointing this out. There was limited data related to the topic and co-occurring conditions in the African population. We do hope that in future, more robust research will emanate from the region.

  • Comment 2:      Fig 1 is the exact figure as Tassone et al. 2023. Even though the proper citation is included, I recommend removing the figure. Authors can include some text elaborating on the impact of the FXPAC across the lifespan.

 Response 2: Agree. We have, accordingly, modified the manuscript. This can be found on pages 4-5, line 93-103

Each disorder in the premutation range is distinct and has its associated clinical characteristics, and may occur over the entire life span of the affected individual.

For example, in infancy, problems with vision and motor functions are encountered; in childhood, it may be anxiety, seizure disorders, neurobehavioural problems, learning issues, social deficits, autism spectrum disorders and ADHD. Young adults may continue with neurobehavioral problems such as anxiety and in addition in mid adulthood endocrine and immune problems may develop such as FXPOI, hypothyroidism, chronic fatigue and fibromyalgia. Also in adulthood, psychiatric and neurological problems may be overt, and there can be significant features of FXAND such as  anxiety, depression, and sleep disorders. The elderly may exhibit neurodegenerative symptoms of  FXTAS – tremors, ataxia, cognitive decline, parkinsonism, autonomic dysfunction and memory problems. 14

Comment 3:  In section 2, epidemiology of FXS in Africa, authors mentioned the prevalence estimate of FXS in the general population and reported on studies in Africa that are mostly based on individuals with intellectual disability. 

It will be helpful to add studies conducted in other countries where the rate is also estimated based on individuals with intellectual disability. It will provide a better picture of similarities/differences in FXS rate between Africa and other parts of the world.

 Response 3: Agree. We have, accordingly, modified the manuscript. See page 7, paragraph 1, line 148 -151

In contrast to studies from Africa, a systematic review and meta- analysis by Hunter et al26 among mostly European and Americans, found that 2.4% (178/7475) of the population with intellectual disability had FXS. This rate is less than all reports from the African subregion, suggesting that FXS may be more prevalent in the region.

Comment 4: The text in section 2.1., epidemiology of FXPAC in Africa, is not relevant to the title of the section. Similar pattern throughout the manuscript. It includes general information about fragile X syndrome or premutation but limited information specific to African populations or studies conducted in Africa (e.g., the section 3, lines 380-453, does not include information related to African population).

 Response 4: Agree. We have, accordingly, modified the manuscript. See page 7, section 2.1, line 144; page 13, paragraph 1, line 283-284.

‘’From our literature search, no specific study addressed FXAND in Africa. This review may serve to create awareness of the disorder in the African region.’’

Comment 5: The structure of the manuscript needs revision. In the introduction section include a brief literature review and then in the main text discuss information related to the African population. 

For example, in the section on "Clinical and Behavioral Phenotypes of FXS", authors can focus more on checklist by Lubala et al. Provide addition information on what is different in the new checklist, what is the implication and how it can be adopted in development of other checklists/ guidelines related to the African population. 

Response 5: Agree. We have, accordingly, modified the manuscript. See pages 16 and 17; lines 348-356

The global prevalence of FXS is estimated at 1 in 5,000 males and 1 in 4,000 -8,000 females.15 A large population screening for FXS among neonates in the United States reported a prevalence rate of 1:5161 among male neonates.16 In Québec, Canada, a screening of over 24,000 neonates found a prevalence rate of 1 in 6,209 among male participants, while FXS was not found among the female neonates screened.17  In a systematic review and meta-analysis by Hunter et al18, the premutation is estimated to occur at a rate of 1 in 250-800 males and 1in 110-270 females, similar to report from a book by Tassone and Hall as editors.19 Most of these findings are from America and Europe.

This new checklist may be utilized albeit cautiously in resource-limited settings as a diagnostic aid for FXS and calls for further studies on clinical and behavioural phenotypes of FXS in Africa, a population with multi-ethno-cultural regions.

Table 2: Lubala et al 39, 91 clinical checklists for FXS

OLD CHECKLIST

NEW CHECKLIST

Not present (score 0)

Present (score 1)

soft and velvety skin with redundancy of skin on the dorsum of hand

soft and velvety skin

flat feet

elongated face*

large and prominent ears

prominent ears

plantar crease

plantar crease

large testes in post pubertal boys

large testicles

Family history of intellectual disabilities

simian crease*

autistic-like behavior

Hyperextensible  joints *

Microcephaly*

Severe intellectual disabilities

Hyperactivity*

short attention span*

autistic-like features

Total score

* = additions to the old checklist

Comments on the Quality of English Language

Comment 6: The manuscript is well written. Here are a few minor suggestions:

 In many cases the sentences are too long or lack clarity. Please revise throughout the manuscript (e.g., Page 1: Lines 37-45, Page 2: Lines 59-61)

Some of the acronyms are defined multiple times. Please revise throughout the manuscript. 

Response 6: Agree. We have, accordingly, revised the manuscript. See page 3, line 62-66; page 4, paragraph 1, line 80-82

There is an inverse relationship between disease severity in FXS females and the activation ratio (AR) meaning, the percentage of cells that have the normal X as the active X). The higher the AR the more FMRP is produced and the higher the IQ.7

Reviewer 2 Report

Comments and Suggestions for Authors

This is a comprehensive review, with a few data on African population, due to the known difficulties cited by the authors

I think the major issue is the lack of new informations, although the efforts of the authors

some minor revisions are:

- some double spaces to be deleted

- line 43-44 (introduction): verb missing

- line 120-121 please rephrase ("persons are unmethylated"?)

- line 124: please change "in fact" with "actually"

- line 175-176 isn't clear, please rephrase

- line 197, I think it's FXPOI and not POI ("33% prevalence of POI") if I correctly understood

- line 216: please rephrase, i.e. "This neurodegenerative disorder usually starts in older carriers above 50 years, and it's more common and more severe in males than females

Comments on the Quality of English Language

-

Author Response

Response to Reviewer 2 Comments

2. Point-by-point response to Comments and Suggestions for Authors

Comments 1: This is a comprehensive review, with a few data on African population, due to the known difficulties cited by the authors

I think the major issue is the lack of new information, although the efforts of the authors

some minor revisions are:

- some double spaces to be deleted

Response 1:  Thank you for pointing this out. We agree with this comment. Therefore, we have deleted the double spaces. The corrections can be found on line 109, page 6, paragraphs 1 and 2; line 114, page 6, paragraph 2; page 7, line 146, paragraph 2; page 8, line 164, paragraph 2; page 9, line 181,paragraph 1; page 11, line 227, paragraph 1; page 13, paragraph 3, line 285; page 14, line  305, paragraph 1; page 17, paragraph 2, line 350; page 18, line 381, paragraph 3;page 21, line 434, paragraph 1; page 22, lines 471 and 473, paragraphs 2 and 3;

Comments 2: - line 43-44 (introduction): verb missing

Response 2: Agree. We have, accordingly, modified the introduction to include the missing verb. See line 64 -65 of the introduction.

Comments 3- line 120-121 please rephrase ("persons are unmethylated"?)

Response 3: Agree. We have, accordingly, rephrased the sentence. ‘’These disorders are in the premutation range, implying that affected persons have unmethylated, CGG repeats in the range of 55 to 200 and the level of FMRP is usually in the normal range.’’ See page 8, paragraph 4, line 149 -151.

Comments 4- line 124: please change "in fact" with "actually"

Response 4: Agree. We have replaced ‘in fact’ with ‘actually’ in the sentence. ‘’They were previously thought to be carriers who were largely unaffected but progressive research over the years has revealed that actually premutation carriers may have a plethora of symptoms and signs that largely cause morbidity and mortality if left unattended.’’ This is on page 7, paragraph 3, line 151 – 154.

Comments 5 - line 175-176 isn't clear, please rephrase.

Reports have shown that delays in the diagnosis of FXPOI occur in 175 women who knew their premutation status and presented with symptoms of POI. This delay  was even more pronounced in women who did not.5

Response 5: Agree. We have, accordingly, rephrased the sentence.

Reports have shown that women who knew their premutation status and also had symptoms of POI experienced delays in getting a diagnosis of FXPOI. However, ‘diagnostic delays’ were more pronounced in women who were unaware of their premutation status.

See page 10, paragraph 1, line 201 – 203

Comments 6: line 197, I think it's FXPOI and not POI ("33% prevalence of POI") if I correctly understood.

Response 6: Agree. We have, accordingly, corrected the sentence.

Essop and Krause21 reported an estimated 33% prevalence of FXPOI (1 in 3 females tested who had a PM) in POI referrals in Johannesburg, South Africa in their 20-year review.

See page 10, paragraph 3, line 221-223.

Comments 7: line 216: please rephrase, i.e. "This neurodegenerative disorder usually starts in older carriers above 50 years, and it's more common and more severe in males than females.

Response 7: Agree. We have, accordingly, rephrased the sentence to ‘This neurodegenerative disorder usually starts in older carriers above 50 years, and FXTAS is more common and more severe in males than females because females have a second unaffected X chromosome that is protective.’ See page 11, paragraph 3, line 241 -242.

Round 2

Reviewer 1 Report

Comments and Suggestions for Authors

Thank you for revising the manuscript. I have no further comments. 

Reviewer 2 Report

Comments and Suggestions for Authors

-

Comments on the Quality of English Language

-